# Temperature-Dependent Viral Pathogenicity: Implications for Attenuation of Viral Vaccines

**DOI:** 10.3390/v18010048

**Published:** 2025-12-28

**Authors:** Kimiyasu Shiraki, Takashi Kawana, Richard J. Whitley

**Affiliations:** 1Hokuriku Health Service Association, Toyama 930-0177, Japan; 2Department of Nursing, Senri Kinran University, Osaka 565-0873, Japan; 3Department of Obstetrics and Gynecology, Teikyo University Mizonokuchi Hospital, Kawasaki 213-8507, Japan; tkawana@med.teikyo-u.ac.jp; 4Division of Pediatric Infectious Disease, Department of Pediatrics, Heersink School of Medicine, University of Alabama at Birmingham, 1720 2nd Ave South, Birmingham, AL 35294, USA; rwhitley@uabmc.edu

**Keywords:** attenuated vaccine, temperature-dependent pathogenicity, viral microlesions, innate immunity, adaptive cellular immunity, temperature-sensitive growth impairment

## Abstract

This review highlights the role of temperature sensitivity, a common feature of attenuated virus vaccines, in mediating attenuation. Viral attenuation mechanisms were analyzed by comparing vaccine characteristics and genes with those of wild-type viruses. Development of varicella vaccines, particularly their attenuation and immunogenicity in immunocompromised children, provided key insights into these mechanisms. Temperature sensitivity leads to smaller viral microlesion formation than wild-type virus by impaired viral replication before recognition by the innate immune system, eliciting protective immunity without causing clinical symptoms. Vaccine candidates were selected based on their attenuation in humans and replication ability for mass production, with impaired growth and temperature sensitivity as common characteristics among many vaccines. Temperature-sensitive rhinoviruses replicate in the nasal mucosa at 33 °C but not in the lungs at 37 °C, demonstrating in vitro and in vivo temperature sensitivity. Similarly, vaccine-induced immunity arises from viral microlesions caused by impaired growth of temperature-sensitive strains; however, these lesions remain small and result in attenuated clinical symptoms. Because of this impaired growth, higher inoculation doses than those of wild-type strains are required to establish infection and elicit immunity. Therefore, clinical attenuation results from impaired viral replication due to temperature sensitivity, yet it induces protective immune responses.

## 1. Introduction

In our previous studies, we reported that, compared with wild-type viruses, temperature-sensitive and temperature-insensitive viral strains exhibit different pathogenicity depending on the temperature at the site of infection [1,2,3]. A literature review on temperature sensitivity in relation to viral attenuation vaccine development was performed (Figure 1). Virus research has expanded significantly through the incorporation of molecular biology and the manipulation of viral genes. Consequently, research using temperature-sensitive viruses has declined since the 1980s. Temperature-sensitive viruses proliferate at 33 °C, similar to wild-type strains, but show decreased proliferative capacity at 37 °C. Therefore, comparisons of the intracellular distribution and localization patterns of mutant gene proteins, as well as complementation of sensitivity mutations, have been conducted at 33 °C and 37 °C for wild-type and temperature-mutant strains, enabling functional analysis of viral genes. Between 1970 and 2000, a high proportion of virology research was dedicated to investigating biological functions and pathogenicity alterations using temperature-sensitive strains. However, once direct analysis of viral gene functions and sequences became possible, analyses using temperature-sensitive strains largely ceased. Since 2000, the proportion of studies on viral temperature sensitivity and pathogenicity has steadily declined. The studies on temperature-sensitive viruses have largely been supplanted by the application of contemporary molecular biology tools to virus research. This review aimed to examine the relationship between infection-site temperature, temperature-dependent viral growth, and immune recognition of viral infection, emphasizing how these factors considerably influence viral pathogenicity.

Between 1972 and 1975, a series of studies on the growth characteristics and animal pathogenicity of experimentally generated temperature-sensitive herpes simplex virus type 2 (HSV-2) were presented as research on temperature-sensitive strains [4,5,6,7]. The pathogenicity of temperature-sensitive HSV-2 strains varied depending on the animal species, infection site, and infection dose, even when the same temperature-sensitive mutant was used for animal infection.

Wild-type and temperature-sensitive HSV-2 isolated from the genital region and herpetic whitlow of a patient with acute myeloid leukemia exhibited temperature-dependent pathogenicity depending on the infection site [1,2]. These viruses retained their pathogenicity in mice. Temperature-sensitive HSV-2 demonstrated pathogenicity in humans and mice, and it was neither nonpathogenic nor attenuated.

Vaccines were developed by creating candidate vaccine strains from wild-type strains using various methods, and attenuated vaccine strains were selected based on their pathogenicity in humans. In this review, temperature sensitivity was identified as a common characteristic among these strains. However, as observed for HSV-2, temperature-sensitive strains do not necessarily become attenuated strains.

Varicella infection in immunocompromised patients is characterized by a prolonged incubation period and severe disease. The Oka varicella vaccine has been established as a safe vaccine for immunocompromised and healthy children, and it is the only varicella vaccine approved by the World Health Organization (WHO). Like other vaccines, the varicella vaccine is temperature-sensitive. Clinical responses in immunocompromised individuals who received the varicella vaccine suggest the attenuation mechanism of temperature-sensitive vaccines. The wild-type virus creates viral microlesions, which are recognized by innate immunity and induce a prodrome characterized by itching and redness, followed by vesiculation and crusting mediated by adaptive cellular and humoral immunity. Temperature-sensitive vaccines, which impair viral replication more at 37 °C than wild-type viruses, form smaller viral microlesions. These are recognized by innate and adaptive immunity; nonetheless, viral microlesions associated with vaccines are too small to cause clinical symptoms, resulting in asymptomatic infection with protective cellular and humoral immunity.

Rhinovirus is a naturally temperature-sensitive virus that replicates efficiently in the nasal cavity at 25–33 °C but not in the lungs, where the temperature is approximately 37 °C. Its pathogenicity is directly influenced by these temperature-dependent replication characteristics, resulting in rhinitis but not pneumonia. In rhinovirus infection, the site of infection and viral replication is limited to the nasal cavity at lower temperatures, and the virus does not replicate in the lungs, demonstrating the infection-site specificity of temperature-sensitive viruses.

Most attenuated vaccine candidates are selected from virus-infected in vitro passaging cultures at 37 °C or another arbitrary temperature, and selected strains are classified based on their properties as highly virulent (comparable to wild-type strains), attenuated, or lacking immunogenicity in humans. Vaccines are selected from attenuated strains and must be capable of sufficient replication for production. Although vaccines were required to be attenuated, immunogenic, and safe, analysis of the attenuation mechanism was not required for approval. As a result, the underlying properties of some vaccines were never thoroughly analyzed. Most vaccine attenuation mechanisms, such as cellular tropism and temperature sensitivity, were investigated using the technology available at the time. The pathogenesis of temperature-sensitive rhinoviruses may be important for understanding temperature sensitivity in vaccine recipients.

Attenuated viral vaccines that exhibit temperature sensitivity include the influenza vaccine (FluMist) [8], Sabin vaccine for poliovirus [9,10], Oka varicella vaccine [11,12,13,14,15,16], infectious bronchitis virus vaccine (IBV) [17,18], yellow fever 17D-MP vaccine [19], measles AIK-C vaccine strain [20,21], mumps virus vaccine [22], parainfluenza virus vaccine [23,24], and respiratory syncytial virus vaccine [25,26]. The ability of these vaccines to induce immunity depends on the minimum infectious dose and the role of innate immunity. Temperature-sensitive strains, which exhibit reduced replication at the injection site and in their target organs, are designed to stimulate an immune response while maintaining low pathogenicity and avoiding overt symptoms.

Various analyses of the attenuation mechanisms of vaccines have identified specific mutations in viral proteins or nucleotide sequences associated with each vaccine. However, how these mutations confer attenuation in vivo remains unclear. Therefore, we conducted a cross-sectional literature review of common mechanisms related to viral attenuation. Compared with wild-type strains, vaccines showed impaired proliferation, although they induced protective immunity without causing clinical symptoms. Notably, temperature sensitivity emerged as a common characteristic among many attenuated vaccines.

## 2. Biological Importance of Temperature Sensitivity of HSV-2 in In Vitro and In Vivo Studies

### 2.1. Pathogenicity of Temperature-Sensitive HSV-2 Mutants in Animals

Temperature sensitivity and host cell specificity were compared among mouse, rabbit, and hamster cells (Table 1) [4,5,6,7]. The replication ability of HSV-2 was 10 times higher in rabbit kidney primary cells (RK cells) than in hamster embryonic fibroblast (HEF) cells. However, as shown in Table 1, compared with the parental strain 333, the temperature-sensitive HSV-2 strains 41, 46, 69, and 74 showed a reduction in replication ability to approximately 1/10,000 at 39 °C, relative to that at 33 °C, in HEF cell lines. In contrast, HSV-2 temperature-sensitive strains replicated at 33 °C in RK cells, similar to the wild-type strain, although their replication ability decreased to approximately 1/10 at 39 °C. Growth reduction for HSV-2 temperature-sensitive strains at 39 °C was 1000-fold greater in HEF cells than in RK cells. Although temperature sensitivity is caused by genetic mutations in the virus, its effect on proliferation depends on the host cells [5,6,7].

HSV-2 temperature-sensitive mutant strains exhibited reduced replication ability in hamster cells compared with that in rabbit cells at 39 °C, and the extent of host cell-mediated inhibition was investigated in animal experiments [4]. When the parental strain was subcutaneously inoculated at 46 plaque-forming units (PFU) per weaned mouse, a 100% mortality rate was observed 14 days post-infection. In contrast, at the same dose, mutant strains 41, 46, 69, and 74 were associated with survival rates of >80%. When the parental strain and temperature-sensitive strains 41, 46, 69, and 74 were subcutaneously inoculated at 4600 PFU per mouse, mortality rates of 100%, 10%, 50%, 85%, and 30%, respectively, were observed at 14 days post-infection. The temperature-sensitive strains exhibited attenuation, although mortality increased with increasing inoculum dose. When the parental strain and mutant strains 41, 46, and 69 were intracerebrally inoculated into weaned mice at 100 PFU, mortality rates of 100%, 40%, 30%, and 0%, respectively, were observed. However, when the parental strain and mutant strains 41, 46, and 69 were inoculated into the brains of mice at 100,000 PFU each, mortality rates of 90%, 100%, 90%, and 50%, respectively, were observed. As the intracerebral inoculation dose of the mutant strains increased, the mortality rate increased to levels comparable to those observed for the parental strain, consistent with the results of subcutaneous inoculation. When the parental strain and mutant strains 46, 69, and 74 were intracerebrally inoculated into hamsters at a dose of 250,000 PFU, 67% of the weaned hamsters died; conversely, the same dose was nonpathogenic in mice, with a 100% survival rate. Although HSV-2 was more virulent in mice than in hamsters when administered intracerebrally, its pathogenicity was lower in hamsters than in mice. Additionally, the pathogenicity of temperature-sensitive strains was lower than that of the parental strain in hamsters. Temperature-sensitive mutant strains exhibited milder replication defects in RK cells at 39 °C than in HEF cells, although the cytopathology associated with infection was more pronounced in the former than in the latter. When the parental strain 333 HSV-2 and mutant strains 46, 69, or 74 were inoculated into the corneas of both eyes of weaned rabbits at 100,000 PFU, the parental strain and mutant strain 41 gradually resulted in keratitis, whereas mutant strains 46, 69, and 74 did not cause keratitis. Temperature-sensitive HSV-2 strains, unlike in HEF cells, exhibited reduced replicative capacity at 39 °C in RK cells while maintaining proliferative capacity. Notably, in corneal infection, all strains except strain 41 showed strongly attenuated virulence, despite the mild degree of temperature sensitivity. This series of studies was conducted between 1972 and 1975, and the mutant genes of temperature-sensitive HSV-2 strains had not yet been determined.

As described above, the pathogenicity of temperature-sensitive HSV-2 strains was influenced by animal species, infection site, and inoculum dose; however, it remains unclear whether this is due to the mutant gene’s function or the temperature-sensitive mutation.

### 2.2. Genital Herpes and Secondary Herpetic Whitlow Caused by Temperature-Sensitive Acyclovir (ACV)-Resistant HSV-2

A 40-year-old man with acute myelogenous leukemia developed genital herpes, which was treated with oral ACV. Despite a 13-week course of oral ACV, his genital lesion increased in size [2]. During this treatment, secondary herpetic whitlow appeared as an ulcerative lesion with vesicles on his right thumb. Despite treatment with oral valacyclovir (1000 mg/day), the whitlow rapidly progressed, eroding and developing crusted lesions with progressive destruction of the fingernail. Six months later, the whitlow on his right thumb recurred, with no recurrence of the genital lesion.

HSV-2 was isolated from the genital and whitlow swabs. The virus isolated from the genital lesion was ACV-sensitive, with an inhibitory concentration (IC_50_) of <1 μg/mL. In contrast, HSV-2 isolates from the first and second whitlow swabs were resistant to ACV, with an IC_50_ of >10 μg/mL. Both whitlow isolates exhibited a cytosine deletion in the four-cytosine stretch at the 819th nucleotide, resulting in a frameshift in the thymidine kinase gene, and a c.566C >T (p.Ala189Val) mutation in the small subunit of ribonucleotide reductase (RR), conferring temperature sensitivity [1,2]. The *RR* mutation of both whitlow isolates was consistent with that of previously described temperature-sensitive HSV strains [1,27,28,29,30]. Both isolates exhibited pronounced impairment of growth at 39 °C compared with 33 °C, with an approximate 50% reduction in plaque formation compared to the genital isolate.

The cutaneous pathogenicity of the genital and two temperature-sensitive whitlow isolates was compared in murine models using midflank skin (33.3 °C) and ear pinnae (27 °C) [2,31,32]. The genital isolate caused skin lesions on the midflank and ear pinna. In contrast, the temperature-sensitive whitlow isolates failed to form lesions at the midflank but caused lesions at the ear pinna. Thus, the temperature-dependent pathogenicity of genital and temperature-sensitive HSV-2 in humans was reproduced in mice.

In conclusion, the temperature-sensitive HSV-2 strain obtained from the patient exhibited temperature-dependent pathogenicity at the site of infection, remained pathogenic, and did not become a nonpathogenic attenuated virus.

### 2.3. Temperature-Sensitive HSV Strains and Attenuation

Temperature-sensitive HSV mutants have been used to study the functions of genes and proteins. HSV mutants that cause temperature sensitivity have been identified in glycoproteins, regulatory proteins, noncoding regions, DNA-binding proteins, alkaline nuclease, and DNA polymerase [33,34,35,36,37,38]. Gene mutations affect the function of proteins and regulatory protein complexes as well as the three-dimensional structure of nucleic acids, leading to reduced function and decreased replication, which result in temperature sensitivity. HSV-2 strains with an RR mutation causing temperature sensitivity showed no pathogenicity when infecting human genitalia and the midflank skin in murine models (33.3 °C); however, they displayed pathogenicity at the thumb as whitlow and the pinna in murine models (27 °C). This showed that, regardless of the gene involved, any mutation that causes temperature sensitivity would result in attenuation at 37 °C but retain pathogenicity at lower temperatures.

The mutation genes of the temperature-sensitive HSV-2 mutants shown in Table 1 have not yet been identified. Although the temperature-sensitive strains, which were possibly carrying different gene mutations, were primarily attenuated, they exhibited varying pathogenicity depending on the animal species, inoculation site, and viral load. Including strains with an RR mutation that were isolated from whitlows, temperature-sensitive strains were fundamentally attenuated compared with wild-type strains. Thus, rather than differences in the gene mutation, temperature sensitivity itself is important for attenuation.

Candidate vaccine strains have been shown to be attenuated when administered via the routes used in clinical studies of the vaccine. Therefore, even temperature-sensitive vaccines do not exhibit pathogenicity in the same manner as that observed in clinical studies.

## 3. Selection of Vaccine Candidates and Temperature Sensitivity

Most attenuated vaccine candidates are selected from virus-infected cultures through in vitro passaging at 37 °C or another arbitrary temperature. The selected strains or fractions are then classified based on their properties as highly virulent strains comparable to wild-type strains, attenuated strains, or strains lacking immunogenicity in humans. The best attenuated vaccine is then selected among them, as shown in Section 5.2, *KMcC varicella vaccine attenuation process* by serial passage [39,40].

Briefly, the KMcC varicella vaccine was examined for its attenuated and immunogenic properties across 10–60 passages, ranging from passages producing varicella symptoms equivalent to those of the wild-type strain to passages showing attenuation with vaccine-level symptoms, and over-attenuation with no clinical symptoms or antibody production, regardless of live virus inoculation. Inoculation with several dozen PFU of low-passage wild-type varicella virus induces viral microlesions that are recognized by innate and adaptive cellular immunity, leading to the development of varicella symptoms and an immune response. In contrast, inoculation with several dozen PFU of the vaccine strain induces viral microlesions with impaired viral growth, which are eliminated by innate immunity, preventing the establishment of infection. Therefore, to induce an immune response, vaccination requires a dose of at least 1000 PFU. This higher dose leads to the formation of numerous viral microlesions that are not eliminated by innate immunity and induce an adaptive immune response; however, no skin lesions develop because viral microlesions are too small. Thus, compared with wild-type strains, the varicella vaccine produces viral microlesions resulting from impaired growth, induces protective immunity, and does not cause clinical symptoms.

Vaccines are selected from clinically attenuated strains that retain sufficient replication ability for mass production. Although vaccines were required to be attenuated, immunogenic, and safe, regulatory approval of the attenuation mechanism itself was not required. Therefore, some vaccines were not analyzed for their properties. Most vaccine attenuation mechanisms, such as altered cellular tropism and temperature sensitivity, were examined using the technology available at the time.

Various analytical methods have been used to investigate the mechanisms underlying attenuated vaccines. When a suitable animal model was available, pathogenicity in animals was also examined. The nucleotide sequences of the wild-type and vaccine strains have been compared, and the proteins and untranslated regions of the poliovirus vaccine have been analyzed in relation to the attenuation [9,10,41,42,43,44,45,46]. However, the attenuation mechanism could not be fully clarified as a common feature of vaccines. Notably, no cross-sectional studies have comprehensively examined attenuation mechanisms across different vaccines to date; nonetheless, temperature sensitivity has been observed as a characteristic common to many vaccines.

Vaccine candidate strains must have sufficient proliferative capacity for large-scale vaccine production and maintain stable properties during culture passages. Temperature-sensitive viruses can arise during viral passages prior to candidate clone selection and may be identified as small plaques among the candidate plaques. These strains retain high proliferative capacity for vaccine production.

## 4. Rhinovirus Infection

Rhinovirus is the primary causative agent of the common cold, leading to symptoms such as rhinorrhea. It is adapted to the nasal temperature range of 25–33 °C [47,48,49]. Rhinovirus replicates efficiently in cells at 33 °C but not at 37 °C, demonstrating characteristic temperature sensitivity. This 4 °C increase in temperature serves as a replication barrier that rhinovirus cannot overcome. Consequently, rhinoviruses proliferate in the nasal mucosa at 25–33 °C, causing rhinitis; however, they do not replicate in the lower respiratory tract at 37 °C, preventing the development of pneumonia [50,51]. We previously isolated rhinovirus from the lungs of a patient with transfusion-associated graft-versus-host disease who had pneumonia [52]. The isolated virus exhibited typical cytopathic effects in human embryonic lung (HEL) cells at 33 °C but not at 37 °C (Figure 2A). The virus was identified as rhinovirus 13, and anti-rhinovirus 13 antibodies were produced by immunization with purified virus. Immunohistochemical analysis localized rhinovirus antigen expression primarily to alveolar type II cells, with no detection in interstitial tissues or alveolar type I cells, which cover approximately 90–95% of the alveolar epithelium (Figure 2B) [52]. Given that the lungs maintain a temperature of 37 °C, the detection of rhinovirus antigens in alveolar type II cells, but not in alveolar type I or interstitial cells, suggests limited viral spread and replication within lung tissue. We speculated that the presence of viral antigens in alveolar type II cells results from virus particles originating in the upper respiratory tract and becoming trapped in these cells, rather than from active replication at 37 °C. The temperature sensitivity of rhinovirus observed in cell culture is consistent with its site of replication and pathogenic expression in humans.

## 5. Varicella Vaccine

### 5.1. Establishment and Characteristics of the Attenuated Varicella Vaccine

Analysis of the attenuated nature of the varicella vaccine and its clinical response revealed that the vaccine shows impaired growth properties and is less proliferative than the wild-type strain, both in cell cultures and in vaccinated children. In immunocompromised individuals, vaccine-induced skin lesions are typically limited to small papules without vesicles, and the rashes tend to be mild and small [53,54,55,56]. In healthy individuals, the varicella vaccine does not usually produce a skin rash; however, tiny papules may appear in some immunocompromised individuals [53,54,55,56]. In healthy individuals, the varicella skin test becomes positive 10–14 days after infection with wild-type varicella, whereas it becomes positive as early as day 4 after varicella vaccination. Thus, subclinical viral microlesions formed by the impaired growth of the varicella vaccine in vivo induce protective adaptive immunity following innate immunity to the virus on day 4, as discussed in Section 6 and Section 7.

Inoculation with several dozen PFU of wild-type varicella virus induces viral lesions large enough to stimulate innate immunity; however, because these lesions are extensive, they are not eliminated by innate immunity alone, resulting in papules, vesicles, and crusts formed through adaptive cellular immunity. In contrast, inoculation with several dozen PFU of vaccine virus induces viral microlesions with impaired viral growth that are eliminated by innate immunity, preventing infection from becoming established. Therefore, when at least 1000 PFU of vaccine is administered, numerous viral microlesions are formed, which are not completely eliminated by innate immunity and subsequently trigger an adaptive humoral and cellular immune response. Lesions remain subclinical due to their small size, with attenuation of the varicella virus reflecting the impaired growth of the vaccine strain.

### 5.2. KMcC Varicella Vaccine Attenuation Process

The KMcC strain of the varicella virus was first isolated in WI-38 cells and incubated for six passages at 32 °C. Thereafter, the virus was passaged in WI-38 cells at 32 °C or 36 °C, but the temperature sensitivity of the KMcC varicella vaccine was not described [39,40]. The KMcC varicella vaccine clearly demonstrated attenuation and immunogenicity in children across successive passages. Early-passage viruses induce skin lesions and immune responses even at low inoculation doses, similar to the wild-type strain. By passage 40, the vaccine with 1150 PFU exhibited attenuated vaccine properties, with few clinical reactions, while still inducing immunity. Beyond passage 50, attenuation progressed such that no clinical symptoms were observed, and the dose effective at passage 40 failed to induce immunity. However, immunity could still be induced by increasing the viral dose without clinical reactions. Specifically, during passages in the 40 s, 115, and 1150 PFU achieved 100% seroconversion. During passages in the 50 s, 650 PFU resulted in 50% seroconversion, whereas increasing the dose to 6500 PFU raised seroconversion to 83%. As the number of passages increased further, neither clinical nor immune responses were detected, even when the viral dose was increased. Because viral replication ability was severely impaired, the virus could not replicate and spread in the body sufficiently to be recognized by innate immunity, resulting in an over-attenuation that failed to induce an immune response, even when large amounts of virus were inoculated.

### 5.3. Oka Varicella Vaccine: Temperature Sensitivity

The Oka varicella vaccine, developed by Dr. Takahashi in 1974 [11,12,13,14,15,16], is used to prevent varicella (chickenpox) and herpes zoster (shingles). Before the development of the varicella vaccine and antiviral agents, varicella caused severe complications in children with acute lymphoblastic leukemia, with a mortality rate of 7–10% [57]. The Oka varicella vaccine has since been safely administered to immunocompromised patients. The Oka strain of the varicella-zoster virus (VZV) underwent 11 passages in HEL cells at 34 °C, followed by 12 additional passages in guinea pig embryo cells. After these passages, the virus demonstrated temperature sensitivity at 39 °C and exhibited increased replication efficiency in guinea pig embryo cells compared with the original and other VZV strains [15,58]. The attenuated Oka varicella vaccine strain displays temperature sensitivity, an altered host range, and modified human cell tropism compared with the parental Oka strain (Table 2 and Table 3). Comparative nucleotide sequence analysis between the original and vaccine-adapted Oka strains identified 42 nucleotide substitutions, including 20 amino acid changes within the 125 kb genome [59]. The vaccine strain demonstrates increased tropism for human umbilical vein and dermal microvascular endothelial cells, while exhibiting reduced replication in neonatal dermal fibroblasts compared with the parental strain [60].

The KMcC and Oka varicella vaccines retain attenuation and immunogenicity up to the 10th passage; however, further passaging results in increased attenuation and reduced immunogenicity, which can be compensated by increasing the viral dose [14,39,40]. The Oka varicella vaccine was developed and has been safely used in immunocompromised individuals, including children with leukemia [16,53,61,62,63]. Compared with the KMcC varicella vaccine, the Oka varicella vaccine is more attenuated and immunogenically stable and is now widely used. The WHO first implemented a seed lot system for the Oka varicella vaccine, maintaining the vaccine virus in a controlled passage system up to the 10th generation [14]. The Oka varicella vaccine is currently used as a WHO-approved varicella vaccine because of its superior immunogenicity, attenuation, and stability compared with the KMcC varicella vaccine.

## 6. Temperature-Sensitive Attenuated Viral Vaccine

### 6.1. Sabin Poliovirus Vaccine: Temperature Sensitivity as an Attenuation Marker

The Sabin poliovirus vaccine has been widely used to prevent poliovirus-related disease and paralysis. This live attenuated vaccine replicates in the human gastrointestinal tract but not in the nervous system. Its attenuation and low neurovirulence correlate with temperature sensitivity at 38–40 °C, in contrast to the optimal replication temperature of 34–37 °C [10,41,42,43,44,45,64,65,66]. Suppression of the temperature-sensitive phenotype is strictly associated with reversion to virulence in nonrecombinant type 3 strains [9]. In addition, the 5′-noncoding region of P1/Sabin enhances the temperature-sensitive phenotype conferred by 3D^pol^ [10,42,43,45]. RNA polymerase and other mutations clearly contribute to temperature-sensitive phenotypes; however, the exact mechanism by which these mutations cause attenuation remains unclear.

### 6.2. Attenuated Temperature-Sensitive Influenza Vaccine (FluMist)

The intranasal influenza vaccine (FluMist) is a temperature-sensitive live influenza virus vaccine that replicates in the nasal mucosa at 33 °C but fails to replicate in the lower respiratory tract, thereby preventing influenza symptoms [8,67,68,69,70,71]. The viral strains used in FluMist are cold-adapted at 25 °C and exhibit temperature sensitivity at 37 °C (Type B strain) or 39 °C (Type A strains) as a result of mutations in the PB1, PB2, and NP genes. Attenuation was demonstrated in ferrets as a model for human influenza infection. The vaccine virus is propagated in the allantoic fluid of specific pathogen-free eggs, which are then harvested, purified, and sterilized. The three influenza strains are combined to create FluMist, with each 0.2 mL intranasal dose containing 10^6.5−7.5^ fluorescent focus units of live attenuated virus, administered equally into both nostrils.

Following nasal administration, vaccine viruses infect and replicate in the nasopharyngeal epithelial cells, inducing an immune response. Viral shedding occurs primarily in nasal secretions, with vaccine viruses being detectable daily for the first 7 days and every other day up to days 25–28. The estimated probability of transmission is 2.4%. Reported side effects include nasal congestion/nasal discharge (59.2%), cough (27.8%), oropharyngeal pain (17.9%), and headache (11.2%), all considerably milder than symptoms of natural influenza infection, which often include fever.

The Centers for Disease Control and Prevention states in their guidance “Live Attenuated Influenza Vaccine (The Nasal Spray Flu Vaccine)” that flu vaccines do not cause influenza illness. The nasal spray vaccine contains weakened (attenuated) viruses, which are cold-adapted and designed to multiply only at cooler temperatures in the nose, not in the lungs or other warmer areas of the body [72]. Thus, temperature-sensitive FluMist, similar to rhinovirus, infects the nasal mucosa and induces an immune response but does not infect the lungs—a common characteristic of temperature-sensitive viruses.

### 6.3. Other Live Attenuated Virus Vaccines and Temperature Sensitivity

The yellow fever 17D-MP vaccine strain showed slight temperature sensitivity at 39.5 °C, with reduced virulence in adult mice [19]. The measles AIK-C vaccine strain, derived from the Edmonston wild-type strain, also exhibited temperature sensitivity at 39 °C [20,21]. Mumps, parainfluenza, and respiratory syncytial virus vaccines exhibited attenuation and temperature sensitivity [22,23,24,25,26].

## 7. Minimal Infectious Dose for Infection

Several temperature-sensitive viruses attenuated in various animal models failed to establish infection when the viral load was low. However, when the infectious dose increased, infection was established and virulence was observed. At very low doses, the replicating virus was eliminated by innate immunity at the site of infection (Figure 3).

The Oka varicella vaccine strain is attenuated and characterized by temperature sensitivity and altered cell tropism compared with the parental Oka strain [12,15,60]. Vaccination with doses exceeding 500–800 PFU within 3 days of household exposure to varicella effectively prevents disease [73]. Thus, the minimal efficacious dose for the varicella vaccine is 1000 PFU/dose [74]. Vaccination with >1000 PFU permits sufficient replication to overcome the eliminating capacity of innate immunity. This, in turn, induces adaptive immunity against the varicella virus, leading to long-lasting protective humoral and cell-mediated immunity. Conversely, infection with <300 PFU results in tiny microlesions and triggers innate immunity at the inoculation site. These innate immune responses eliminate the viral microlesion, and protective adaptive cellular and humoral immunity is not induced, even though the live virus is inoculated and replicates. Vaccination with <300 PFU of the Oka varicella vaccine failed to induce protective immunity and did not protect against varicella.

The KMcC varicella vaccine strain has demonstrated attenuation and immunogenicity compared with the wild-type strain, with over-attenuation observed through serial passages. At 10 passages, inoculation with 70 PFU resulted in varicella in 82% of patients, with 100% seroconversion [39,40]. Following 40 passages, 115 PFU induced rash in 62% of patients with 100% seroconversion. Inoculation with 6500 and 650 PFU of 50-passaged viruses resulted in no clinical symptoms, although seroconversion rates were 83% and 50%, respectively. Before attenuation (10 passages), the wild-type virus at 70 PFU caused rash in 82% of patients and vesiculation in 46%, with 100% infection. Even at 50 passages, 6500 PFU established infection in 83% of recipients. Seroconversion with the attenuated virus at 50 passages decreased to 50% at 650 PFU but recovered to 83% at 6500 PFU. Therefore, an attenuated virus requires a higher dose to establish infection.

Natural varicella infection occurs via the respiratory tract or eyes. The incubation period is approximately 14 days. Viral dissemination throughout the body occurs via viremia from 5 days before clinical symptom onset until 1 day after symptom onset. The rate of virus isolation from blood is the highest at 1 to 2 days before symptom onset [47]. During the incubation period, viral microlesions spread, and innate immunity induces erythema in the prodromal phase. Subsequently, acquired cellular immunity induces vesicles and antibody responses. Intradermal reactions are detectable when the rash appears, and antibody responses are typically detectable around 2 days later.

In immunocompromised patients, the innate immune response is delayed, prolonging the incubation period. This leads to larger viral microlesions, resulting in more severe rashes, internal complications, and, occasionally, death. Because the immune response in immunocompromised individuals is delayed and causes severe disease, vaccines other than the Oka varicella vaccine are contraindicated.

The attenuated Oka varicella vaccine requires a high infectious dose for the induction of immunity and does not cause clinical symptoms following vaccination. The virus cannot be isolated from the blood or pharynx of vaccinated children 3 to 14 days post-vaccination; this suggests low-level viral replication in vaccine recipients [47]. Cell-mediated immunity, assessed by skin reactivity, appears 4 to 5 days after vaccination, followed by the emergence of antibodies.

## 8. Importance of Innate Immunity for Controlling Viral Infection Preceding Adaptive Immunity

Langerhans cells play a critical role in the innate immune responses of the skin by detecting replication of HSV and VZV. These cells secrete various cytokines and host factors, orchestrating the influx of inflammatory cells (Figure 3). Such inflammatory reactions can create a prodrome and ultimately induce adaptive humoral and cell-mediated immune responses [75,76,77]. Innate immunity contributes to the manifestation of erythema, itching, tingling, and allodynia during the prodromal phase [78,79,80]. Adaptive cell-mediated immune responses then result in the formation of characteristic skin lesions, including papules, vesicles, and crusts. The importance of innate immunity is exemplified by the “photodistribution” vesicles observed after ultraviolet (UV) light exposure. UV irradiation impairs Langerhans cell function, diminishes innate immunity, and thereby exacerbates skin lesions (Figure 3C) [81]. As shown in Figure 3, worsening of HSV skin lesions (Figure 3D) and varicella (Figure 3E) is observed 3 days after UV exposure. Lesions in sun-exposed areas are denser and more severe because UV light damages Langerhans cells, whereas lesions in shielded areas are sparser and milder, reflecting intact innate immunity. Innate immunity inhibits or eliminates local viral replication in the early stages of infection. Consequently, fewer viral lesions are observed in shaded areas. Although viral microlesions form at similar densities in shaded and sun-exposed skin, those in sunburned areas progress unchecked as a result of impaired innate immunity, resulting in a dense skin rash. In shaded areas, however, viral microlesions are eliminated by intact innate immune responses (Figure 3B). When innate immunity is fully functional, viruses, viral antigens, and viral microlesions are eliminated. In infections caused by temperature-sensitive viruses, replication remains localized at the inoculation site. Viral microlesions with impaired growth are removed by innate immunity in the early stages, preventing infection and the induction of adaptive immunity, such as skin lesions and antibody production [82,83]. Therefore, a large vaccine dose is required to overcome the innate immune barrier. Such vaccination induces both innate and adaptive immunity without producing clinical symptoms, as viral microlesions remain too small to generate clinically apparent lesions.

**Figure 3 viruses-18-00048-f003:**
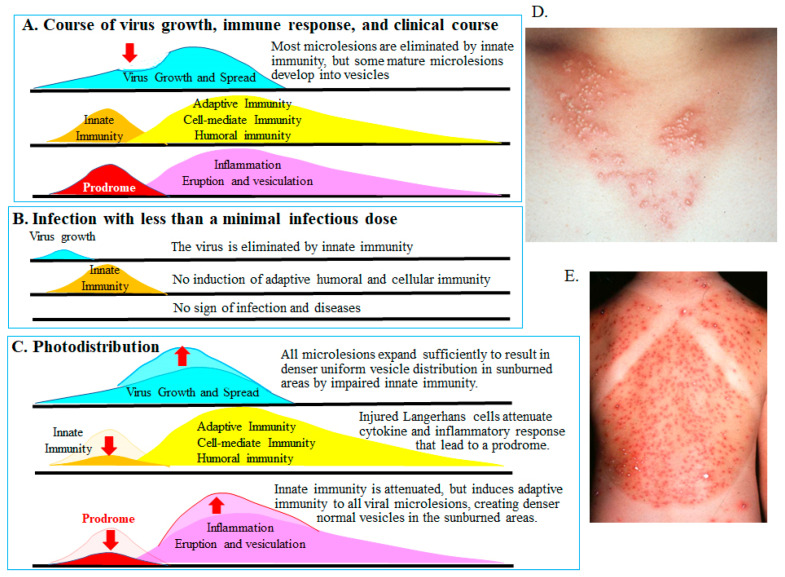
Innate and acquired immunity and photodistribution [75,76,77,81]. The figure illustrates the relationship between viral replication and innate and adaptive immune responses in HSV and VZV infection. (**A**). Transition from innate to adaptive immunity. Langerhans cells recognize viral replication of HSV and VZV in the skin, initiating innate immunity by secreting cytokines and interferons and recruiting inflammatory cells. These responses suppress early viral replication (downward red arrow) and limit viral spread during the prodrome. Subsequently, locally accumulated immunocompetent cells induce antigen-specific adaptive immunity—cellular and humoral—against virus-infected cells, leading to the formation of characteristic vesicular skin lesions. (**B**). Infection below the minimal infectious dose. In cases of low viral load or attenuated growth caused by temperature-sensitive strains, innate immunity eliminates the limited viral microlesions. Consequently, adaptive responses—humoral and cellular—are not induced, and no clinical symptoms appear. (**C**). Sunburn (ultraviolet exposure) impairs Langerhans cell function, inhibiting innate immunity and prodromal suppression (downward red arrows). As a result, even small viral microlesions that would normally be eliminated can replicate and disseminate. Subsequent adaptive cell-mediated immunity leads to an exacerbated and dense distribution of HSV or VZV skin lesions in the sunburned area, a pattern known as photodistribution (upward red arrows). (**D**). A woman in her 20 s developed vesicles distributed consistently over the anterior cervical region exposed to sunlight 2 days earlier. Vesicles and erythema are denser and more uniform in the sun-exposed area compared with shaded areas, reflecting photodistribution. The sparse lesion distribution in shaded areas results from the elimination of virus-infected cells by intact innate immunity (**B**), highlighting its role in clearing viral microlesions. (**E**). An 8-year-old girl presented with varicella and fever 3 days after swimming in the sea. Vesicles clustered densely in the sunburned area (photodistribution by VZV). In contrast, eruptions were sparse in the shaded areas protected by shoulder straps, as innate immunity eliminated most viral microlesions and prevented lesion formation (**B**). Sunburned and shaded skin harbor a similar distribution of viral microlesions before the prodrome. As shown in (**B**,**C**), intact innate immunity (shaded areas) eliminates viral microlesions, preventing lesions, whereas impaired innate immunity (sunburned areas) allows numerous skin lesions to form. Thus, innate immunity plays a crucial role in reducing viral lesion burden by eliminating small infected foci (viral microlesions) before adaptive immunity is induced. Dr. Yasumoto provided the images. The authors obtained permission from MDPI to reuse the figure under the Creative Commons CC BY license [70].

## 9. Significance of Temperature in Viral Pathogenesis in Vaccines

Although attenuation of vaccines has been investigated individually and differences from wild-type strains have been clarified, the mechanism underlying vaccine attenuation remains unclear. Cross-sectional studies on attenuated vaccines have not been conducted to date, and the characteristics common to all vaccines remain undefined. Among the various distinctions between wild-type viruses and vaccine strains, we have identified temperature sensitivity as a common feature of vaccine attenuation.

Replication of vaccines with temperature sensitivity is impaired at the human body temperature of 37 °C. Viral microlesions are recognized by innate immunity and induce adaptive cellular immunity without clinical symptoms because the impaired viral microlesions are too small to cause symptoms. Temperature-sensitive FluMist replicates in the nasal mucosa without replicating in the upper respiratory tract or lungs, by a mode of infection similar to that of rhinovirus infection, and produces protective immunity against influenza. Cold-adapted and temperature-sensitive IBVs were attenuated in chickens, similar to the IBV [17]. The interaction between tiny viral microlesions created by temperature-sensitive vaccines and innate immunity is essential for attenuation in vivo and the induction of protective immunity.

## 10. Discussion

This review focused on the attenuation of vaccines and their relationship with temperature sensitivity, impaired growth, recognition of viral microlesions by innate and adaptive immunity, and attenuation of clinical symptoms. Before the year 2000, the relationship between viral temperature sensitivity and virulence or attenuated virulence was well established, and vaccine development was actively progressing, as evidenced by a substantial number of publications (Figure 1).

Many attenuated vaccines show temperature sensitivity. However, as demonstrated by the temperature-dependent pathogenicity of temperature-sensitive HSV-2, temperature-sensitive strains are not necessarily attenuated. Although genetic mutations have been extensively studied in poliovirus vaccines, the mechanism underlying attenuation has not been clarified through viral genetic mutations. Clinical symptoms observed in immunocompromised patients receiving varicella vaccines highlighted the importance of innate immunity-mediated virus recognition and immune responses, as demonstrated by photodistribution, for the attenuation of live viruses. Although the mechanism of temperature sensitivity in vaccines is virus-specific, the common feature is impaired proliferation at the human body temperature of 37 °C. In other words, innate immunity-mediated recognition results in small viral microlesions resulting from impaired viral proliferation, allowing cellular and humoral immunity to be induced without causing clinical symptoms because viral microlesions are too small.

Rhinovirus infection is associated with acute exacerbations in patients with chronic obstructive pulmonary disease (COPD); however, the status of the virus in the lungs has not been reported. We have observed a patient with rhinovirus pneumonia in whom viral antigens were not detected in alveolar type I cells, which represent approximately 90–95% of the lung interstitium and alveolar epithelium. Instead, they were detected in alveolar type II cells, which do not support viral replication at 37 °C. In patients with COPD, acute exacerbations may be related to inflammation caused by dispersion of the virus into the airway epithelium through nasal secretions and its entrapment by alveolar type II cells. Rhinovirus multiplies in the nasal mucosa at 33 °C and causes rhinitis, but does not multiply in the lungs at 37 °C, indicating that the temperature characteristics of temperature-sensitive viruses in vitro are also reflected in vivo.

A comparison of vaccines with wild-type strains showed that vaccine-specific mutations involved genes related to transcriptional regulation and nucleic acid synthesis, although no common target genes were identified across different vaccines. A common characteristic of the vaccines was growth impairment resulting from the temperature sensitivity at 37 °C in cell culture, while still maintaining the ability to proliferate for vaccine production. Such minimal growth impairments were caused by mutations such as amino acid substitutions in viral proteins or mutations in gene regulatory regions, resulting in impaired proliferation at 37 °C. Various vaccine-specific analyses revealed that, compared to wild-type strains, growth was suppressed at 37 °C but maintained at 33 °C. The growth impairment common to vaccines was temperature sensitivity.

Table 4 shows temperature-sensitive mutations associated with attenuation in the Sabin 634 poliovirus vaccine, FluMist influenza vaccine, Oka varicella vaccine, measles vaccine 635 (AIK-C), and mumps vaccine (Urabe). As seen with HSV, temperature-sensitive mutation 636 genes are not limited to specific genes with common functions. Interactions between tem-637 perature-sensitive mutations related to attenuation and other gene mutations result in im-638 paired growth common to the vaccines, leading to attenuation in humans. 

Temperature sensitivity at the cellular level is reproduced in vivo, as observed in rhinovirus infections. Compared to the wild-type strain, the growth impairment of the Oka varicella vaccine in cell culture conferred protective immunity without varicella symptoms in otherwise healthy vaccinated children. However, after Oka varicella vaccination, mild skin lesions were observed in immunocompromised individuals, such as children with leukemia. The elimination and recognition of proliferated viral microlesions of the Oka varicella vaccine by innate immunity in vivo can be better understood through the photodistribution of varicella in patients (Figure 3). As discussed in Section 4 regarding the varicella vaccine, viral microlesions arise from impaired growth associated with the temperature-sensitive vaccine virus, which induces innate and adaptive protective immunity. Viral microlesions successfully induce protective immunity but are too small to cause clinical symptoms, resulting in asymptomatic or mild infections. Therefore, a higher dose of the vaccine is required to establish infection than wild-type virus and acquire protective immunity. Similar to the IBV, cold-adapted and temperature-sensitive IBV is attenuated at 41 °C in chickens and can induce protective immunity [17]. The spectrum of vaccine gene mutations was found to be largely temperature-sensitive, resulting in attenuated biological responses.

## 11. Conclusions

Although each virus is thought to have an attenuation mechanism, these changes are insufficient to fully explain attenuation. A common feature of vaccines is growth impairment. The history of varicella vaccine development revealed that the attenuation mechanism was primarily impaired growth, mainly by temperature sensitivity, along with the subsequent immune and clinical responses in vivo. Viral microlesions with impaired growth induce immune responses, although clinical symptoms do not occur because the viral microlesions are too small to cause clinical manifestations. The vaccine has reduced replication capacity, although sufficient replication is maintained for vaccine production. These findings suggest that genetic mutations in vaccines are phenotypically temperature-sensitive mutations. This new concept of vaccine attenuation mechanisms will provide important information for improving current vaccines and developing new ones.

## Figures and Tables

**Figure 1 viruses-18-00048-f001:**
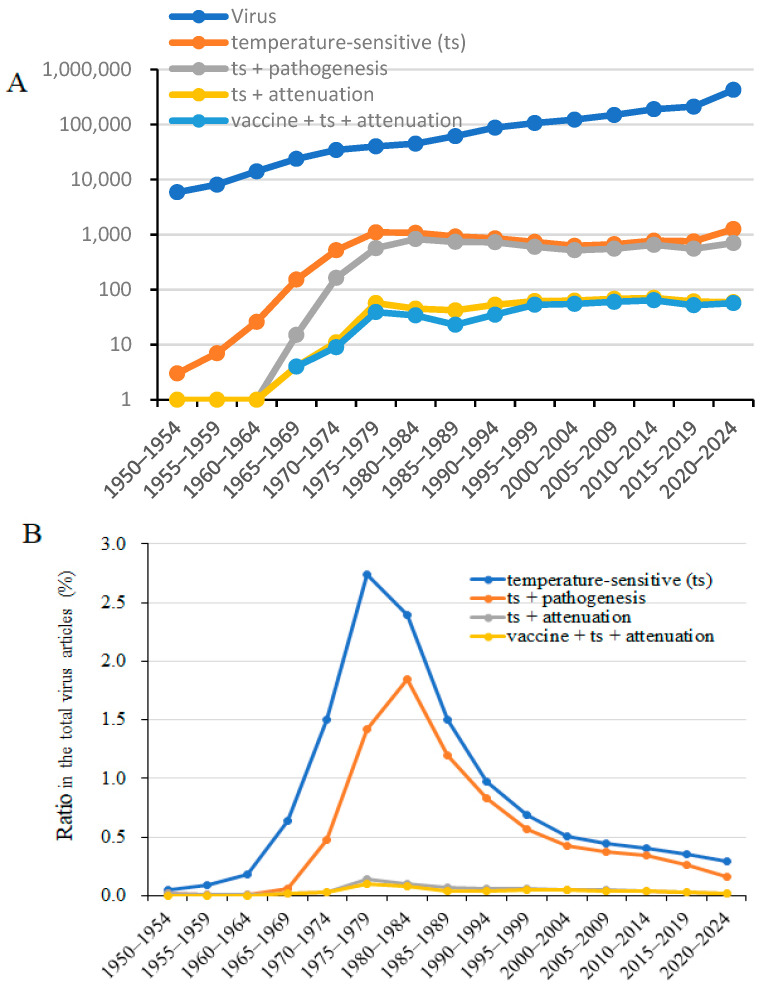
Chronological count of studies on viruses, including “temperature-sensitive (ts),” “ts + pathogenesis,” “ts + attenuation,” and “virus vaccines + ts + attenuation,” since 1950. (**A**). Number of papers on “temperature-sensitive,” “attenuation,” and “vaccine + ts + attenuation” in chronological order from PubMed as of 5 January 2025. (**B**). Proportion of papers specifically on “temperature-sensitive (ts),” “ts + pathogenesis,” “ts + attenuation,” and “vaccines + ts + attenuation” within virus research.

**Figure 2 viruses-18-00048-f002:**
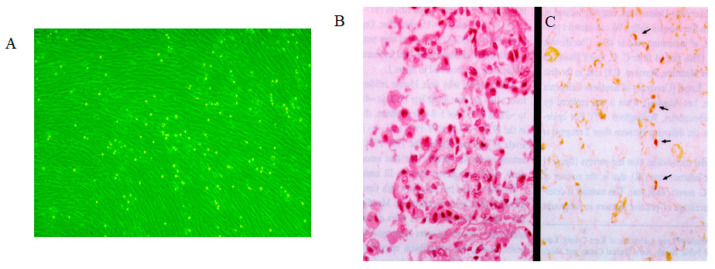
Rhinovirus cytopathology and immunohistochemistry in rhinovirus pneumonia. (**A**). Typical cytopathology (rounding) of rhinovirus infection in human embryonic lung cells at 33 °C. (**B**). Histopathologic analysis of lung tissue from an infant with rhinovirus pneumonia shows thickening of the alveolar septum, hyperplasia of the alveolar lining cells, and macrophages in the alveoli (hematoxylin–eosin stain; original magnification, ×350). (**C**). Analysis with hyperimmune serum (modified avidin–biotin peroxidase complex method) reveals localization of rhinovirus antigen in the cytoplasm of hyperplastic alveolar lining cells *(arrows),* desquamating swollen alveolar epithelial cells, and macrophages in the alveoli (original magnification, ×175). Permission to reuse this figure was obtained from Oxford University Press [52].

**Table 1 viruses-18-00048-t001:** Viral growth and pathogenicity of parental and temperature-sensitive HSV-2 mutants [4,5,6,7].

		Parental Strain 333	Mutant Strain 41	Mutant Strain 46	Mutant Strain 69	Mutant Strain 74
Viral growth in cultures(PFU)	Hamster embryo fibroblasts 39 °C/33 °C	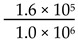	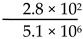	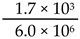	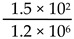	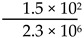
Rabbit kidney cells at 39 °C	2.8 × 10^5^	7.0 × 10^4^	8.5 × 10^4^	2.9 × 10^4^	3.5 × 10^4^
Viral pathogenicity in animals(% mortality)	Subcutaneous inoculation (Mouse) 46 PFU	100	<20	<20	<20	<20
Subcutaneous inoculation (Mouse) 4600 PFU	100	10	50	85	30
Intracranial inoculation (Mouse) 100 PFU	100	40	30	0	
Intracranial inoculation (Mouse) 100,000 PFU	90	100	90	50	
Intracranial inoculation (Hamster) 250,000 PFU	67		0	0	0
Rabbit keratitis 100,000 PFU	++	+	−	−	−

PFU: plaque-forming units. Compared with the HSV-2 parental 333 strain, viral growth of HSV-2 temperature-sensitive mutants in hamster embryo fibroblasts was lower by 1000–10,000 fold at 39 °C than at 33 °C, indicating temperature sensitivity. Viral growth in rabbit kidney cells at 39 °C was less strongly suppressed than in hamster embryo fibroblasts. Mortality rates (%) are shown following subcutaneous or intracranial inoculation of mice and hamsters. Keratitis was induced in rabbits by corneal inoculation.

**Table 2 viruses-18-00048-t002:** Temperature sensitivity of the Oka varicella vaccine and wild-type strains of varicella-zoster virus (VZV) [58].

Strain	Number of Passages in HEL Cells	Infectivity at the Indicated Temperature (PFU/0.2 mL)			Infectivity Ratio
		39 °C	37 °C	34 °C	39 °C:34 °C
Oka vaccine	-	4.0 × 10^1^	1.5 × 10^3^	1.5 × 10^3^	1:38
Oka original	7	3.0 × 10^2^	8.5 × 10^2^	9.5 × 10^2^	1:3.2
Izawa	7	7.0 × 10^1^	4.0 × 10^2^	4.0 × 10^2^	1:5.7
Inoue	5	3.0 × 10^2^	1.0 × 10^2^	1.0 × 10^3^	1:3.3
Kawaguchi	12	2.0 × 10^2^	6.5 × 10^2^	7.0 × 10^2^	1:3.5

HEL: human embryonic lung cells; PFU: plaque-forming units. The authors obtained permission from Oxford University Press to reuse this table [58].

**Table 3 viruses-18-00048-t003:** Differences in cell tropism between the original Oka and attenuated Oka varicella vaccine strains [40].

Strain	HEL	Umbilical Endothelium	Dermal Endothelium	Dermal Fibroblast	Hepatocyte
Oka original	100	4.1 ± 2.4 *	11.7 ± 2.1	6.6 ± 2.5 *	714 ± 212
Vaccine	100	17.3 ± 13.2 *	26.0 ± 3.7	3.9 ± 1.7 *	777 ± 335
*p* value		<0.001 (*n* = 4)	0.001 (*n* = 3)	<0.001 (*n* = 5)	0.261 (*n* = 6)

HEL: human embryonic lung cells. The parental Oka original and Oka vaccine strains were inoculated into cells in six-well plates, and the number of plaques was counted using a binocular microscope. Plaque numbers are expressed as the mean percentage ± S.D. of that observed in HEL cell cultures in each set of experiments. The virus titers of the Oka original and Oka vaccine strains were 4.45 × 10^3^ and 2.78 × 10^4^ PFU/mL, respectively. *p* values indicate the statistical differences in the plaque formation between the Oka original and Oka vaccine strains by two-way ANOVA, with the number of experiments shown in parentheses. A *p* value of <0.05 was considered statistically significant and is indicated by an asterisk (*). The authors obtained permission from Oxford University Press to reuse this table [60].

**Table 4 viruses-18-00048-t004:** Mutations of viral vaccines related to attenuation and temperature sensitivity.

Vaccine	Mutations Related to Attenuation and Temperature Sensitivity
Sabin Poliovirus Vaccine	A Y73H change in viral RNA polymerase (3Dpol), and the 5′ nontranslated region (NTR) and coding sequences of the capsid proteins [46]. nucleotides 935 of viral protein (VP) 4, 2438 of VP3, and 2795 and 2879 of VP1 and the 3′-noncoding region, 3Dpol [10]. The 91st amino acid of VP3 [41]. A 3Dpol codon (C-7071-->U, Thr-362-->Ile) and the 3′ noncoding region of the genome (A-7441-->G) [42].
FluMist (influenza vaccine)	Basic polymerase 1 (PB1) (K391E, E581G, A661T), PB2 (N265S), Nucleoprotein (NP) (D34G) of the master donor virus-A strain [71].
Oka Varicella Vaccine	Compared with Oka parent virus: 42 base substitutions resulting in 20 amino acid conversions in open reading frames 6 (component of DNA helicase–primase complex), 9A (envelope protein gN), 10 (tegument protein), 21 (tegument protein), 31 (envelope glycoprotein gB), 39 (major capsid protein), 50 (envelope glycoprotein gM), 52 component of helicase–primase complex), 55 (helicase), 59 (uracil-DNA glycosylase), 62 (transcription regulator), and 64 (a fusion modulatory factor) [59]. Temperature-sensitive mutation and attenuation mutation have not been identified.
Measles Vaccine (AIK-C)	The Phosphoprotein (P439-Pro) [21].
Mumps Vaccine (Urabe strain)	Hemagglutinin-neuraminidase (HN) 468, Large protein (L) 722 in Domain III of the viral RNA-dependent RNA polymerase, a noncoding substitution at nucleotide 9307 (L290), and a substitution at nucleotide 15289 in the untranslated region (UTR) of the genome [22].

## Data Availability

No new data were created or analyzed in this study.

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
