# Peer review of "Viruses2026, 18(1), 48;https://doi.org/10.3390/v18010048"

_viruses, 2025, doi:10.3390/v18010048_

Round 1

Reviewer 1 Report

Comments and Suggestions for Authors

This review discusses the role of temperature sensitivity in viral attenuation and proposes a unifying phenotypic framework linking impaired viral replication at higher body temperatures to the formation and innate immune clearance of small viral microlesions. The topic is timely, and the manuscript is generally well organized and clearly written. The cross-virus comparisons make the central argument accessible and coherent.

The main issue needing clarification is the treatment of temperature-sensitive HSV-2 strains, which remain pathogenic despite impaired growth. A explicit explanation of why temperature sensitivity results in attenuation for some viruses but not for HSV-2 would strengthen the argument.

Overall, this is a solid and coherent review that only needs minor revision.

Author Response

Responses to the comments raised by Reviewer 1

Thank you for reviewing and commenting on our manuscript. We have revised our manuscript in response to your valuable comments.

The previous Reviewer-Editor has argued that attenuation of temperature-sensitive strains is well known, so temperature sensitivity in vaccines isn’t novel. In response, we added a section on HSV-2 to clarify that, although temperature-sensitive strains are attenuated compared to wild-type strains, they can still cause disease at low temperature sites in mice and humans.

The final vaccine candidate strain is the selected one that shows no pathogenicity and exhibits the highest immunogenicity among the candidate strains. The vaccine strains discussed in this manuscript all shared the common trait of temperature sensitivity. The vaccine was approved based on factors such as attenuation, stability through passage, and safety without toxicity. Before the 1980s, identifying the mutated gene responsible for attenuation was not considered essential for vaccine development, as molecular technology to detect mutations in the attenuation gene was unavailable. In response to the Reviewer-Editor’s comment, a section on HSV-2 temperature sensitivity was added to the manuscript.

Comment on the temperature-sensitivity and pathogenicity of HSV-2

“The main issue needing clarification is the treatment of temperature-sensitive HSV-2 strains, which remain pathogenic despite impaired growth. A explicit explanation of why temperature sensitivity results in attenuation for some viruses but not for HSV-2 would strengthen the argument.”

As mentioned in the manuscript, temperature sensitivity is defined by the reduced replication capacity at 33°C compared to 37°C, although this difference varies among strains. In a temperature-sensitive HSV-2 study reported in Table 1, they observed differences in pathogenicity at the inoculation site, with different inoculation doses, and among different animal species. The genetic mutations in these strains remain unidentified, and the relationship between specific gene mutations and pathogenicity has not been analyzed and is still unclear.

We demonstrated that although temperature-sensitive strains isolated from humans did not cause lesions in the patient's genital area and in mouse midflank skin, they were pathogenic on the patient's low-temperature thumb and mouse ear pinnae. Thus, by infecting the low-temperature ear skin in mice—an area not typically the site of experimental infection—we showed that temperature-sensitive strains can exhibit pathogenicity at low-temperature sites in the ear pinnae but not at midflank temperature. Since the mutation in this temperature-sensitive strain is in ribonucleotide reductase, we believe that the local temperature at the infection site determines the strain's pathogenicity, rather than the mutation in the specific gene.

  As seen with temperature-sensitive HSV-2 isolated from humans, even attenuated vaccine strains may exhibit pathogenicity at low temperature, depending on the inoculation site, route of administration, and viral load. Vaccines with temperature sensitivity would not exhibit pathogenicity, because the administration route was selected to avoid any pathogenicity in the clinical study.

As suggested by the Reviewer 1, we added Section 2.3 to the manuscript.

A explicit explanation of why temperature sensitivity results in attenuation for some viruses but not for HSV-2 would strengthen the argument.

According to the reviewer's comment, we have inserted the section “2.3. Temperature-Sensitive HSV Strains and Attenuation” into the revised manuscript as follows.

2.3. Temperature-Sensitive HSV Strains and Attenuation

Temperature-sensitive HSV mutants have been used to study the functions of genes and proteins. HSV mutants that cause temperature sensitivity have been identified in glycoproteins, regulatory proteins, non-coding regions, DNA-binding proteins, alkaline nuclease, and DNA polymerase [33-38]. Gene mutations affect the function of proteins and regulatory protein complexes as well as the three-dimensional structure of nucleic acids, leading to reduced function and decreased replication, which result in temperature sensitivity. HSV-2 strains with an RR mutation causing temperature sensitivity showed no pathogenicity when infecting human genitalia and the midflank skin in murine models (33.3°C); however, they displayed pathogenicity at the thumb as whitlow and the pinna in murine models (27°C). This showed that, regardless of the gene involved, any mutation that causes temperature sensitivity would result in attenuation at 37°C but retain pathogenicity at lower temperatures.

The mutation genes of the temperature-sensitive HSV-2 mutants shown in Table 1 have not yet been identified. Although the temperature-sensitive strains, which were possibly carrying different gene mutations, were primarily attenuated, they exhibited varying pathogenicity depending on the animal species, inoculation site, and viral load. Including strains with an RR mutation that were isolated from whitlows, temperature-sensitive strains were fundamentally attenuated compared with wild-type strains. Thus, rather than differences in the gene mutation, temperature sensitivity itself is important for attenuation.

Candidate vaccine strains have been shown to be attenuated when administered via the routes used in clinical studies of the vaccine. Therefore, even temperature-sensitive vaccines do not exhibit pathogenicity in the same manner as that observed in clinical studies.”

References 33–38

  1. Purifoy, D. J.; Powell, K. L. Herpes simplex virus DNA polymerase as the site of phosphonoacetate sensitivity: temperature-sensitive mutants. J Virol 1977, 24, (2), 470-7. DOI:10.1128/JVI.24.2.470-477.1977.
  2. Hafner, J.; Mohammad, F.; Farber, F. E. Alkaline nuclease activity in cells infected with herpes simplex virus type 1 (HSV-1) and HSV-1 temperature-sensitive mutants. Biochim Biophys Acta 1987, 910, (1), 85-8. DOI:10.1016/0167-4781(87)90097-2.
  3. Dixon, R. A.; Sabourin, D. J.; Schaffer, P. A. Genetic analysis of temperature-sensitive mutants which define the genes for the major herpes simplex virus type 2 DNA-binding protein and a new late function. J Virol 1983, 45, (1), 343-53. DOI:10.1128/JVI.45.1.343-353.1983.
  4. Addison, C.; Rixon, F. J.; Palfreyman, J. W.; O'Hara, M.; Preston, V. G. Characterisation of a herpes simplex virus type 1 mutant which has a temperature-sensitive defect in penetration of cells and assembly of capsids. Virology 1984, 138, (2), 246-59. DOI:10.1016/0042-6822(84)90349-0.
  5. DeLuca, N. A.; Courtney, M. A.; Schaffer, P. A. Temperature-sensitive mutants in herpes simplex virus type 1 ICP4 permissive for early gene expression. J Virol 1984, 52, (3), 767-76. DOI:10.1128/JVI.52.3.767-776.1984.
  6. DeLuca, N.; Person, S.; Bzik, D. J.; Snipes, W. Genome locations of temperature-sensitive mutants in glycoprotein gB of herpes simplex virus type 1. Virology 1984, 137, (2), 382-9. DOI:10.1016/0042-6822(84)90230-7.

Reviewer 2 Report

Comments and Suggestions for Authors

The manuscript titled, “Temperature-Dependent Viral Pathogenicity: Implication for Attenuation of Viral Vaccines,” is a review of literature regarding the role of temperature in attenuating vaccines. The focus of the manuscript is primarily on viral replication and vaccine dose. The authors present a substantial survey of select vaccines, their growth/manufacture conditions, and their provenance in the literature. The manuscript is well written and provides logical and complete references; however there is a significant gap present in this review with respect to genetics and the mechanisms by which a virus becomes attenuated that requires the authors attention.

General Comment: The authors mention in the abstract and several times throughout the manuscript about comparing genes; however, there are no genetic changes mentioned specifically in the entirety of the manuscript. Perhaps the authors should consider adding this analysis or discussion of the genetic changes for the reader to understand the genetic events brought on by temperature-based attenuation and thus the mechanism(s) of attenuation.

Lines 48-50: The authors state that gene function analysis has decreased since the year 2000 but do not discuss why this might be. Perhaps this is because genetic changes in the population or escape mutations make this analysis hard to consider. Furthermore, Figure 1a supports that the total number of temperature sensitive publications has not decreased, only the ratio as shown in Figure 1b. The change in normalized data may not be substantial considering the number of publications in the field of virology and addition of new frontiers, viruses, outbreaks, etc. The authors should at least address the change in ratio in terms of the changing body of literature. For example, a search for “human T-cell lymphotropic virus-type III” in pubmed yields only 166 results while a search for without quotations of human T-cell lymphotropic virus-type III or HIV yields 449,878 results. This change in ratio might suffer from a similar change in terminology. Another example is the expansion of literature topics like a search for HIV diagnosis after the invention of the PCR assay would make the antibody or other methods of diagnosis seem like a changed ratio but would still incorporate HIV diagnostics.

Line 77-79: The authors state that temperature sensitivity does not necessarily become attenuated strains but the mechanism underlying this might be similar to mentioned in the previous comment (e.g. genetic changes). The authors should discuss the mechanisms of attenuation, specifically the changes observed in the genes and the phenotypic outcomes.

Line 81-83: The authors mention the Oka vaccine as the only one approved by the WHO, but this is the basis for the FDA licensed varicella vaccine as well, where being immunocompromised is a contraindication. This difference between WHO and FDA products might be an interesting talking point, especially if there is any literature out there discussing the differences.

Author Response

Responses to the comments raised by Reviewer 2

Thank you for reviewing and commenting on our manuscript. We have revised our manuscript in response to your valuable comments.

.

General Comment: The authors mention in the abstract and several times throughout the manuscript about comparing genes; however, there are no genetic changes mentioned specifically in the entirety of the manuscript. Perhaps the authors should consider adding this analysis or discussion of the genetic changes for the reader to understand the genetic events brought on by temperature-based attenuation and thus the mechanism(s) of attenuation.

Temperature-sensitive mutants of HSV have been used to study the functions of genes and proteins. As a result, mutations in HSV that cause temperature sensitivity have been identified in glycoprotein (1), regulatory protein (2), non-coding regions (3), DNA-binding protein (4), alkaline nuclease (5), and DNA polymerase (6). These mutations are believed to make the virus temperature-sensitive because temperature affects the three-dimensional structures and functions of proteins and nucleic acids, as well as the hydrogen bonds between molecules and the regulatory protein complexes. This results in structural changes in proteins and nucleic acids, leading to reduced function and decreased replication at 37°C compared with 33°C.

  1. Genome locations of temperature-sensitive mutants in glycoprotein gB of herpes simplex virus type 1. DeLucaet al. Virology 1984;137:382-9.
  2. Temperature-sensitive mutants in herpes simplex virus type 1 ICP4 permissive for early gene expression. DeLuca et al., J Virol. 1984;52:767-76.
  3. Characterisation of a herpes simplex virus type 1 mutant which has a temperature-sensitive defect in penetration of cells and assembly of capsids. Addison et al. Virology 1984;138:246-59.
  4. Genetic analysis of temperature-sensitive mutants which define the genes for the major herpes simplex virus type 2 DNA-binding protein and a new late function. Dixonet al. J Virol. 1983;45:343-53.
  5. Alkaline nuclease activity in cells infected with herpes simplex virus type 1 (HSV-1) and HSV-1 temperature-sensitive mutants. Hafneret al. Biochim Biophys Acta 1987;910:85-8.
  6. Herpes simplex virus DNA polymerase as the site of phosphonoacetate sensitivity: temperature-sensitive mutants. Purifoy et al. J Virol. 1977;24:470-7.

In Section 2.2, HSV-2 strains with a ribonucleotide reductase mutation causing temperature sensitivity showed no pathogenicity when infecting the mouse midflank (33.3°C), but displayed pathogenicity similar to the wild-type strain when infecting the ear pinnae (27°C). This showed that, regardless of which gene is involved, any mutation causing temperature sensitivity will lead to attenuation at 37°C but retain pathogenicity at low temperatures. The mutant gene responsible for temperature-sensitive strains has yet to be identified. Therefore, while the relationship between the mutant gene and pathogenicity remains unclear, temperature-sensitive strains commonly show attenuation. Therefore, we have inserted the section “2.3. Temperature-Sensitive HSV Strains and Attenuation” into the revised manuscript to explain the relationship between temperature-sensitive mutation and attenuation as follows.

2.3. Temperature-Sensitive HSV Strains and Attenuation

Temperature-sensitive HSV mutants have been used to study the functions of genes and proteins. HSV mutants that cause temperature sensitivity have been identified in glycoproteins, regulatory proteins, non-coding regions, DNA-binding proteins, alkaline nuclease, and DNA polymerase [33-38]. Gene mutations affect the function of proteins and regulatory protein complexes as well as the three-dimensional structure of nucleic acids, leading to reduced function and decreased replication, which result in temperature sensitivity. HSV-2 strains with an RR mutation causing temperature sensitivity showed no pathogenicity when infecting human genitalia and the midflank skin in murine models (33.3°C); however, they displayed pathogenicity at the thumb as whitlow and the pinna in murine models (27°C). This showed that, regardless of the gene involved, any mutation that causes temperature sensitivity would result in attenuation at 37°C but retain pathogenicity at lower temperatures.

The mutation genes of the temperature-sensitive HSV-2 mutants shown in Table 1 have not yet been identified. Although the temperature-sensitive strains, which were possibly carrying different gene mutations, were primarily attenuated, they exhibited varying pathogenicity depending on the animal species, inoculation site, and viral load. Including strains with an RR mutation that were isolated from whitlows, temperature-sensitive strains were fundamentally attenuated compared with wild-type strains. Thus, rather than differences in the gene mutation, temperature sensitivity itself is important for attenuation.

Candidate vaccine strains have been shown to be attenuated when administered via the routes used in clinical studies of the vaccine. Therefore, even temperature-sensitive vaccines do not exhibit pathogenicity in the same manner as that observed in clinical studies.”

Concerning Fig. 1

Lines 48-50: The authors state that gene function analysis has decreased since the year 2000 but do not discuss why this might be. Perhaps this is because genetic changes in the population or escape mutations make this analysis hard to consider. Furthermore, Figure 1a supports that the total number of temperature sensitive publications has not decreased, only the ratio as shown in Figure 1b. The change in normalized data may not be substantial considering the number of publications in the field of virology and addition of new frontiers, viruses, outbreaks, etc. The authors should at least address the change in ratio in terms of the changing body of literature.

I agree with Reviewer 2's point that the explanation for “Why has research on temperature-sensitive strains decreased?” is lacking. Temperature-sensitive mutants were the primary tools to analyze viral characteristics and functions during the 1970s and 1980s. Animal experiments on temperature-sensitive strains of HSV-2 were conducted in 1972 and 1975 (Table 1), but the mutations responsible for their temperature sensitivity remain unidentified. The poliovirus vaccine (about 7.5 kb) has been used worldwide since the 1950s; its temperature sensitivity was discovered in 1979, and its molecular mechanisms of temperature sensitivity and neurovirulence genes were subsequently studied in the 1990s.

The Mesh keywords of “temperature-sensitive,” “temperature-dependent,” and “temperature-related words” were examined in a PubMed search, and “temperature-sensitive” was the best word among the examined keywords.

From the 1970s to the early 1980s, research primarily focused on analyzing viral properties in cell culture, using viral temperature sensitivity. Molecular biological analysis methods such as Southern blotting and Northern blotting were employed, but nucleotide sequencing was not yet widespread. Subsequently, it became possible to determine nucleotide sequences, express proteins, and intentionally introduce genetic mutations into viral proteins, enabling the analysis of traits resulting from these mutations. With the introduction of these research methods, the necessity for using temperature-sensitive strains, which had been one of the mainstream approaches for analyzing viral characteristics, gradually diminished. Applying current molecular technology standards to vaccine development prior to the 1980s is inappropriate. Understanding this background explains why analysis of genes related to vaccine attenuation was not a regulatory requirement. This is the reason why genetic analysis of vaccines has not been performed to this day.

We have inserted “Virus research has expanded significantly through the incorporation of molecular biology and the manipulation of viral genes. Consequently, research using temperature-sensitive viruses has declined since the 1980s.” in the Introduction in the revised manuscript.

Comment on Oka varicella vaccine and the WHO

“Line 81-83: The authors mention the Oka vaccine as the only one approved by the WHO, but this is the basis for the FDA licensed varicella vaccine as well, where being immunocompromised is a contraindication. This difference between WHO and FDA products might be an interesting talking point, especially if there is any literature out there discussing the differences.”

Regarding the Oka varicella vaccine and the WHO, the Requirements for varicella vaccine (live) in WHO Expert Committee on Biological Standardization 725 (1985) and 848 (1994), describe guidelines for the virus seed lot system for varicella vaccines and for vaccine production. Dr. Takahashi, developer of the Oka varicella vaccine; Dr. Hilleman, of Merck Sharp & Dohme Research Laboratories, developer of the KMcC varicella vaccine; and Dr. Quinnan, of the FDA, participated in the meeting. Thus, the Oka varicella vaccine became the only varicella vaccine recognized by the WHO. Every country follows the guidelines of the seed lot system and vaccine production and produces Oka varicella vaccine. Implementation details, such as vaccine approval criteria, target age groups, and vaccination timing, are determined by each country's regulatory authorities based on local conditions. As reviewers noted, decisions on target populations and contraindications are left to each country. Consequently, even in countries where the varicella vaccine is approved, the recommended age for vaccination varies by nation. One dose of Oka varicella vaccine contains at least 1,500 PFU  in the USA and 1000 PFU in Japan. Immunocompromised individuals are generally included, but specific contraindications also differ by country.

Therefore, a detailed discussion of contraindications requires a comparison of the package inserts of each nation. WHO defined the varicella vaccine strain and the vaccine production process, but the WHO products do not exist. The FDA determined the age groups eligible for the vaccine, the timing of administration, and contraindications.
